# Chemical Composition and Protective Possibilities of Juglans Nigra Leaves and Green Husks Extracts: DNA Binding and Micronucleus Assay in Human Lymphocytes

**DOI:** 10.3390/plants13121669

**Published:** 2024-06-16

**Authors:** Katarina M. Rajković, Miroslava Stanković, Milan Markićević, Gordana Zavišić, Sanja Vranješ-Đurić, Drina Janković, Zorica Obradović, Dalibor Stanković

**Affiliations:** 1The Academy of Applied Preschool Teaching and Health Studies, Kosančićeva 36, 37000 Kruševac, Serbia; cpo@vinca.rs; 2Nuclear Facilities of Serbia, Mike Petrovića Alasa 12-14, Vinča, 11001 Belgrade, Serbia; 3Institute for Oncology and Radiology of Serbia (IORS), Department of Radiotherapy Physics, Pasterova 14, 11000 Belgrade, Serbia; 4Faculty of Pharmacy, University Business Academy Novi Sad, Trg mladenaca 5, 21101 Novi Sad, Serbia; gordana.zavisic@faculty-pharmacy.com; 5“Vinča” Institute of Nuclear Sciences—National Institute of the Republic of Serbia, University of Belgrade, Serbia, POB 522, 11001 Belgrade, Serbiacentar@vinca.edu.rs (Z.O.); 6Faculty of Chemistry, University of Belgrade, Studentski Trg 12-16, 11000 Belgrade, Serbia

**Keywords:** *J. nigra* extracts, cyclic voltammetry, DPPH, micronucleus assay, chemical composition

## Abstract

To better understand the mechanism of action of the compounds in the ethanolic extracts of *J. nigra* leaves and green husks, their binding to CT-DNA was investigated. This study was conducted to elucidate the in vitro protective effect of extracts against chromosomal damage in mitogen-induced human lymphocytes and investigate the possible application of selec+ted extracts as a natural source of polyphenolic compounds. Using HPLC-MS analysis, 103 different compounds were identified as having a higher number of active species, which is consistent with their activity. The frequency of micronuclei (MN) was scored in binucleated cells, and the nuclear proliferation index was calculated. Cyclic voltammetry experiments demonstrate that the nature of the interaction between extracts and CT-DNA is a synergy of electrostatic and intercalative modes, where leaves extracts showed a higher ability to bind to DNA. Extracts showed excellent antioxidant activity. At a concentration of only 4 µg/mL, extract of *J. nigra* leaves and the green husks reduced the incidence of MN by 58.2% and 64.5%, respectively, compared to control cell cultures.

## 1. Introduction

Free radicals, according to their definition, can exist independently and represent a molecular species that contains an unpaired electron. Their main characteristic is that they are very unstable compounds and, therefore, very reactive. Looking at their structure, their reaction with other particles can be two-fold: accepting electrons or giving electrons to that particle, so they act as oxidizing and reducing agents [1]. By definition, free radicals are a very broad group of compounds, but the most important group is oxygen-containing radicals (hydroxyl radical, superoxide anion radical, hydrogen peroxide, singlet oxygen, hypochlorite, nitrogen oxide radical, and peroxynitrite radical). Their appearance is related to a change in the state of the organism, and since they have high reactivity, very often in interaction with membranes and cells, they can damage compounds that are essential for the normal functioning of cells [2].

Environmental agents that cause mutations can lead to the creation of tiny nuclei in the liquid inside cells during a specific stage of cell life. These tiny nuclei can come from parts of chromosomes that lack a specific region or entire chromosomes that cannot move correctly with the others during a different stage of cell division. Monitoring chromosome harm in human populations is commonly performed by studying these tiny nuclei in certain blood cells. The method of using the cytokinesis-block technique allows for a precise count of these tiny nuclei in cells that have completed a certain stage of division, making it a dependable indicator of chromosome harm [3].

The human body has an innate ability to mend DNA injury; however, excessive damage can significantly impact overall health. Extensive DNA damage has the potential to disrupt cellular processes such as glucose metabolism, protein distribution, and cell replication. Antioxidants, which are stable molecules capable of giving an electron to and neutralizing harmful free radicals, help reduce their ability to cause damage. These antioxidants mainly prevent cellular damage by scavenging free radicals [4]. Herbal extracts can improve antioxidant status and reduce oxidative stress in humans. Varieties of medicinal plants are recognized as a source of natural antioxidants that can protect from oxidative stress and thus play an important role in the chemoprevention of diseases [5,6]. In contrast to conventional medications that may result in adverse physical and occasionally mental side effects, herbal remedies have demonstrated minimal side effects or possibly no drawbacks at all.

Black walnut (*J. nigra*) is a type of walnut that has a long history of use, dating back to ancient times, as a source of wood and for its delicious nuts. Black walnut kernels are known to offer many health benefits because they contain many beneficial components such as tannins, essential fatty acids linoleic acid, oleic acid, palmitic acid, stearic acid, and linolenic acid (omega-3), as well as minerals such as magnesium and potassium. However, apart from the kernels, other parts of *J. nigra* are also used in traditional medicine and are thought to provide various health benefits [7]. Walnuts harbor bioactive components like phytosterols and α-linolenic acid, which could potentially have anti-cancer properties. These properties are exhibited through the activation of apoptosis by influencing the expression of apoptotic genes and thwarting proliferation. Moreover, walnuts are known to elevate the expression of tp53 genes, Bax, and caspase-3, along with their respective biomolecules, in cancerous cells [8]. Although research does not support the use of black walnut in the prevention or treatment of health conditions, black walnut extract has long been used in herbal medicine. Black walnut oil is utilized for addressing digestive issues. Nutritionally comparable to English walnuts, black walnuts are highly regarded for their robust, earthy taste and impressive nutrient content. They have been associated with numerous health advantages, such as reducing the risk of heart disease and infections and aiding in weight loss. The outer shells, or hulls, of black walnuts are packed with antioxidant and antibacterial compounds, making them beneficial for naturally combating parasitic and bacterial infections. This paper aims to explore the advantages and potential safety considerations associated with black walnuts. Additionally, a study revealed that green walnut husk contains high levels of polyphenols, which exert antioxidant and antimicrobial effects [9]. The husks contain antioxidants and are used in extracts and supplements for medicinal purposes, such as to treat parasitic infections or decrease inflammation [10]. Walnuts offer a rich reserve of antioxidants, which are capable of stalling or retarding cellular harm induced by volatile molecules referred to as free radicals. Moreover, the husks of black walnuts possess distinct antibacterial characteristics and are harnessed in the preparation of herbal medicinal extracts and supplements. These husks are particularly rich in tannins, which are known for their antibacterial attributes [11]. A test-tube study found that black walnut husk extracts have antioxidant and antibacterial activities, preventing the growth of Staphylococcus aureus, a bacteria that can cause infections [9].

Black walnut extract is commonly incorporated into wormwood complex supplements due to its potent antibacterial characteristics. The wormwood complex tincture is a blend made from black walnut husks, wormwood plants, and cloves, serving as a natural solution against parasitic infections. Some individuals utilize this extract as a mouthwash to eliminate oral bacteria [12]. Moreover, extracts derived from black walnut leaves have therapeutic properties for treating skin conditions like eczema, psoriasis, and warts. Historically, Native Americans have utilized black walnut tree leaves to address issues such as diarrhea, bile-related problems, and intestinal cramps [13]. Additionally, the husk extract acts as a natural dye for hair, skin, and apparel owing to its tannins that naturally darken surfaces. The antimicrobial actions of walnut husk extracts extend to combating Gram-positive bacteria [14,15]. Furthermore, these extracts can inhibit xanthine oxidase, an enzyme linked to hyperuricemia, an inflammation-inducing metabolic disorder associated with gout [16]. High in naphthoquinones, walnut green husks offer a spectrum of potential applications. Notably, juglone, a naphthoquinone present in walnuts, has exhibited cytotoxic effects against cultured melanoma cells [17].

There is a need to study the biological properties of the compounds in *J. nigra* leaves and the green husks. Therefore, this work aims to investigate electrochemically the interaction of the compound from *J. nigra* leaves and the green husks with DNA using cyclic voltammetry (CV) and to evaluate their protective effects on peripheral blood human lymphocytes of a healthy donor. However, further investigations are required to assay the antioxidant effect in vivo and to evaluate its relevance to human health.

## 2. Results and Discussions

UHPLC-Orbitrap MS characterization of walnut extracts resulted in the detection of 103 compounds in total (Table 1). The identified compounds could be divided into seven different groups: (1) hydroxybenzoic acid derivatives (28 compounds); (2) hydroxycinnamic acid derivatives (14 compounds); (3) flavonoid glycosides (29 compounds); (4) flavonoid aglycones (14 compounds); (5) quinones (10 compounds); (6) fatty acids (4 compounds); and (7) 4 compounds classified as other metabolites. In general, it can be said that the leaf extract is richer in terms of the number of identified compounds than the husk extract, which was expected. Figure 1 shows the chromatograms of the base peaks of the examined extracts, where it can be seen that their metabolic profiles differ significantly. Among hydroxybenzoic acids, gallic acid derivatives were the most detected, and quercetin and myricetin derivatives dominate among flavonoid glycosides. Derivatives of quinones, specific for Juglans species, were found in both tested samples, with the exception of 1,4-naphthalenedione hexoside, which was not detected in the husk extract.
plants-13-01669-t001_Table 1Table 1LC/HRMS data for metabolites identified in walnut extracts.NoCompound Name*t*_R_, minMolecular Formula,[M–H]^–^Calculated Mass,*m*/*z*Exact Mass,*m*/*z*Δ mDaMS^2^ Fragments, (% Base Peak)HuskLeafHydroxybenzoic acid derivatives







1HHDP-hexose0.54C_20_H_17_O_14_^−^481.06238481.059662.72249.03931 (4), 257.0087 (2), 275.01849 (67), 300.99759 (100), 481.05991 (12)−✚2Galloyl hexose0.59C_13_H_15_O_10_^–^331.06707331.065511.56125.02389 (9), 168.00566 (9), 169.01352 (71), 211.02385 (74), 241.03458 (7), 271.04474 (100)✚✚3Gallic acid0.61C_7_H_5_O_5_^–^169.01425169.013390.86125.02388 (100), 169.01350 (47)✚✚4Galloyl-HHDP-hexose1.26C_27_H_21_O_18_^–^633.07334633.069923.41169.0134 (9), 275.01837 (23), 300.99747 (100)–✚5*p*-Hydroxybenzoic acid2.33C_7_H_5_O_3_^–^137.02442137.023840.5893.03412 (100), 137.02374 (47)✚✚6Dihydroxybenzoyl hexose2.78C_7_H_5_O_4_^–^153.01933153.018720.61108.02118 (13), 109.02905 (100), 153.01872 (50)✚–7Digalloyl hexose5.13C_20_H_19_O_14_^–^483.07803483.075262.77125.02384 (18), 169.01346 (94), 271.04474 (100), 313.05499 (45), 331.06573 (24), 483.07538 (14)✚–8Di-gallate5.15C_14_H_9_O_9_^–^321.02521321.023641.57125.02381 (11), 158.03659 (12), 169.01343 (100)✚–9Methyl-galloyl hexose5.28C_14_H_17_O_10_^–^345.08272345.080442.28125.02382 (14), 151.00325 (10), 169.01347 (100)✚✚10Ethyl-galloyl hexose5.37C_15_H_19_O_10_^–^359.09837359.095912.46124.01598 (9), 151.00311 (16), 168.00560 (7), 169.01338 (26), 197.04462 (15), 359.09616 (100)✚✚11Dimethyl ellagic acid6.00C_16_H_9_O_8_^–^329.03029329.029091.20271.02356 (16), 299.01831 (24), 314.04178 (100), 329.02805 (33)✚✚12Urolithin M_5_6.00C_13_H_7_O_7_^–^275.01973275.018181.54229.01335 (5), 231.02907 (4), 257.00781 (14), 275.01831 (100)–✚13Tri-galloyl-hexose6.03C_27_H_23_O_18_^–^635.08899635.085563.43169.01344 (100), 211.02388 (12), 271.04465 (10), 295.04465 (11), 313.05508 (88), 465.06509 (72)✚✚14Ethyl gallate6.10C_9_H_9_O_5_^–^197.04555197.044441.11124.01591 (30), 125.02377 (13), 168.00548 (11), 169.01334 (39), 197.04443 (100)✚✚15Ellagic acid pentoside6.11C_19_H_13_O_12_^–^433.04125433.038582.67299.98962 (61), 300.99738 (100), 433.03857 (23)–✚16Galloyl-coumaroyl hexose6.20C_22_H_21_O_12_^–^477.10385477.101132.72151.00302 (20), 163.03934 (50), 169.01352 (100), 301.03394 (44), 313.05511 (85), 477.10129 (26)✚–17Ellagic acid6.30C_14_H_5_O_8_^–^300.99899300.997321.67300.99753 (100)✚✚18Digalloy-feruloyl hexose I6.42C_30_H_27_O_17_^–^659.12537659.122043.33169.01331 (100), 211.02367 (30), 271.04453 (66), 313.0549 (22), 423.05460 (22), 483.07538 (51)✚✚19Ellagic acid galloyl pentose6.61C_26_H_17_O_16_^–^585.05221585.049342.87299.98926 (15), 300.99741 (100), 433.03864 (54)–✚20Digalloy-feruloyl hexose II6.91C_30_H_27_O_17_^–^659.12537659.121923.46169.01364 (4), 271.04468 (100), 331.06561 (8)✚✚214-O-Galloyl-chlorogenate7.47C_23_H_21_O_13_^–^505.09877505.095842.92135.04457 (16), 161.02388 (4), 173.04469 (43), 179.03410 (100), 191.05516 (100), 353.08551 (14)✚–22Galloyl-cinnamoyl hexose7.53C_22_H_21_O_11_^–^461.10894461.106212.72125.02386 (19), 147.04451 (27), 151.00304 (53), 161.06003 (22), 169.01344 (100), 211.02380 (20)✚✚23Urolithin C8.04C_13_H_7_O_5_^–^243.02990243.028691.21171.04427 (17), 199.03981 (16), 215.03362 (39), 243.02887 (100)✚✚24Galloyl deoxypentose8.43C_12_H_13_O_8_^–^285.06159285.060141.46124.01600 (6), 125.02383 (21), 168.00587 (3), 169.01340 (100), 285.06027 (15)✚✚255-O-Galloyl-chlorogenate8.88C_23_H_21_O_13_^–^505.09877505.095982.79135.04448 (11), 161.02402 (4), 173.04463 (6), 179.03413 (76), 191.05522 (100), 353.08551 (11)✚✚26Ethyl digallate8.89C_16_H_13_O_9_^–^349.05651349.054192.32169.01363 (2), 197.04449 (100)✚✚27*p*-Hydroxybenzyl-malonic acid9.01C_10_H_9_O_5_^–^209.04555209.044421.1392.02625 (15), 93.03411 (4), 136.01601 (6), 137.02373 (19), 165.05481 (100)✚✚28Methyl ellagic acid9.79C_15_H_7_O_8_^–^315.01464315.013191.45227.03426 (3), 241.05029 (4), 256.07297 (4), 299.98959 (100), 300.09918 (5), 315.01309 (29)✚✚*Hydroxycinnamic acid derivatives*







291-O-Caffeoylquinic acid1.06C_16_H_17_O_9_^–^353.08781353.085332.47135.04459 (27), 179.03415 (73), 191.05525 (100)✚–30Caffeoyl hexose I4.95C_15_H_17_O_9_^–^341.08781341.085612.19135.04448 (24), 161.02371 (36), 177.05484 (21), 179.03410 (100), 221.04437 (45), 281.0654 (17)✚–31Esculetin5.06C_9_H_5_O_4_^–^177.01933177.018430.91115.05496 (4), 133.02888 (4), 159.04437 (35), 175.0392 (14), 177.01924 (100)✚✚32Caffeoylshikimic acid I5.68C_16_H_15_O_8_^–^335.07724335.075691.55135.04459 (45), 137.02382 (13), 155.03444 (9), 161.02371 (87), 173.04427 (4), 179.03410 (100)✚✚335-O-Caffeoylquinic acid5.70C_16_H_17_O_9_^–^353.08781353.085402.40135.04459 (27), 179.03415 (73), 191.05525 (100)✚–344-O-Feruloylquinic acid5.92C_17_H_19_O_9_^–^367.10346367.101082.37134.03656 (9), 137.02383 (5), 149.05981 (4), 155.03456 (6), 173.04471 (100), 193.04970 (26)✚–35*p*-Coumaric acid6.05C_9_H_7_O_3_^–^163.04007163.039140.93119.04967 (100), 163.03929 (14)✚✚36Caffeoyl hexose II7.10C_15_H_17_O_9_^–^341.08781341.085482.32135.04456 (22), 161.02373 (29), 179.03413 (100), 221.04456 (41), 251.05498 (10), 281.06537 (25)✚–37Ferulic acid7.16C_10_H_9_O_4_^–^193.05063193.049621.0193.03409 (11), 121.02881 (8), 121.06521 (4), 149.06004 (100), 175.03938 (12), 193.04961 (16)–✚38Hydroxy-methyl coumarin7.57C_10_H_7_O_3_^–^175.04007175.039001.07131.04953 (6), 175.03928 (100)✚✚394-O-Caffeoylquinic acid8.41C_16_H_17_O_9_^–^353.08781353.085402.40135.04459 (32), 173.04474 (100), 179.03416 (75), 191.05524 (53)✚–40Caffeoylshikimic acid II8.86C_16_H_15_O_8_^–^335.07724335.075541.70135.04456 (51), 137.02386 (8), 155.03407 (3), 161.02371 (48), 173.04446 (4), 179.03410 (100)✚✚415-O-Feruloylquinic acid8.88C_17_H_19_O_9_^–^367.10346367.101042.41134.03677 (27), 149.06021 (9), 155.03441 (4), 173.04466 (6), 193.04971 (100)✚–42Trimethoxycoumarin9.52C_12_H_11_O_5_^–^235.06120235.059971.23119.04974 (9), 163.03987 (9), 177.05469 (100), 191.03426 (10)✚✚*Flavonoid glycosides*







43Myricetin 3-O-[2″-(ethyl-galloyl)]-rhamoside5.96C_30_H_27_O_16_^–^643.13046643.128342.12151.00316 (4), 169.01369 (11), 178.99771 (5), 197.04471 (10), 316.02106 (100), 317.02820 (24)✚–44Myricetin 3-O-hexoside-7-O-rhamnoside6.02C_27_H_29_O_17_^–^625.14102625.137833.19151.00325 (2), 178.99808 (5), 316.02103 (100), 317.02866 (59), 463.08588 (60)–✚45Myricetin 3-O-rhamnoside6.25C_21_H_19_O_12_^–^463.08820463.085242.96178.99792 (3), 316.02090 (100), 317.02863 (19)✚✚46Myricetin 3-O-pentoside6.29C_20_H_17_O_12_^–^449.07255449.072360.19135.02936 (10), 151.00291 (4), 199.03865 (6), 287.05582 (9), 316.02100 (100), 317.02841 (16)✚✚47Myricetin 3-O-rhamnoside-7-O-hexoside6.49C_27_H_29_O_17_^–^625.14102625.137613.41316.02066 (100), 317.02832 (97), 463.08521 (74), 464.09094 (2), 478.07269 (99), 479.08002 (50)✚✚48Taxifolin 3-O-pentoside6.49C_20_H_19_O_11_^–^435.09329435.090702.58125.0238 (23), 151.00302 (100), 178.99736 (15), 273.03864 (7), 285.03906 (64), 303.04251 (9)✚✚49Quercetin 3-O-pentoside6.52C_20_H_17_O_11_^–^433.07764433.074702.94300.02618 (100), 301.03384 (45)✚✚50Quercetin 3-O-rhamnoside6.63C_21_H_19_O_11_^–^447.09329447.090462.82151.00291 (4), 178.99786 (4), 255.02876 (3), 271.02267 (3), 300.02609 (100), 301.03384 (67)✚✚51Quercetin 3-O-hexoside-7-O-rhamnoside6.72C_27_H_29_O_16_^–^609.14611609.142883.23300.02609 (17), 301.03397 (100), 447.09103 (19), 462.07803 (89), 463.08493 (21)✚✚52Kaempferol 3-O-pentoside (Juglanin)6.83C_20_H_17_O_10_^–^417.08220417.080631.57101.02406 (6), 113.02426 (5), 255.02856 (10), 284.03116 (100), 285.03851 (28)✚✚53Myricetin 3-O-(2″-vanniloyl)-rhamnoside6.89C_29_H_25_O_15_^–^613.11989613.116863.03178.99765 (3), 316.02069 (100), 317.02817 (18), 433.12637 (12)✚✚54Myricetin 3-O-(2″-galloyl)-rhamnoside6.90C_28_H_23_O_16_^–^615.09916615.095883.28151.00298 (4), 178.99774 (10), 316.02118 (6), 317.02859 (100)✚✚55Myricetin 3-O-hexoside6.97C_21_H_19_O_13_^–^479.08311479.080142.97151.00307 (2), 316.02063 (100), 317.02805 (14)✚✚56Quercetin 3-O-(2″-vanniloyl)-rhamnoside7.15C_29_H_25_O_14_^–^597.12498597.121463.52151.00340 (5), 177.05467 (28), 178.99800 (7), 271.02271 (6), 300.02594 (100), 301.03381 (56)✚✚57Myricetin 3-O-(2″-*p*-hydroxybenzoyl)-rhamnoside7.36C_28_H_23_O_14_^–^583.10933583.106832.50195.06500 (12), 285.03873 (20), 316.02057 (100), 317.02817 (25)✚✚58Quercetin 3-O-(2″-sinapoyl)-rhamnoside7.42C_32_H_29_O_15_^–^653.15119653.148452.74151.00290 (7), 178.99768 (8), 223.06029 (7), 271.02347 (8), 300.02606 (100), 301.03381 (79)✚–59Taxifolin 3-O-(6″-galloyl)-hexoside7.50C_28_H_25_O_16_^–^617.11481617.112821.99125.02384 (6), 169.01346 (55), 273.03922 (31), 285.03906 (64), 303.049530 (100), 455.05960 (86)✚–60Isorhamnetin 3-O-rhamnoside7.52C_22_H_21_O_11_^–^461.10894461.106442.49145.02905 (45), 314.04196 (100), 315.05108 (41)✚✚61Quercetin 3-O-methyl-hexuronide7.53C_22_H_19_O_13_^–^491.08311491.080842.27175.03891 (8), 271.05014 (28), 300.02609 (100), 301.03381 (15), 447.09042 (24)✚✚62Myricetin 3-O-[2″-(methyl-galloyl)]-rhamoside7.65C_29_H_25_O_16_^–^629.11481629.112082.73169.01384 (13), 178.99789 (12), 183.02931 (7), 316.02115 (73), 317.02881 (100), 331.04623 (7)–✚63Myricetin 3-O-(2″-*p*-coumaroyl)-hexoside7.68C_30_H_25_O_15_^–^625.11989625.116803.09151.00291 (4), 178.99771 (11), 179.03271 (3), 316.02100 (46), 317.02866 (100), 463.08618 (7)✚✚64Quercetin 3-O-(2″-*p*-coumaroyl)-rhamnoside7.84C_30_H_25_O_13_^–^593.13007593.126643.43151.00284 (4), 178.99783 (4), 300.02600 (100), 301.03366 (56), 429.08151 (2), 447.09085 (5)✚✚65Kaempferol 3-O-(2″-*p*-coumaroyl)-rhamnoside8.06C_30_H_25_O_12_^–^577.13515577.132452.70119.04954 (4), 145.02881 (9), 163.03926 (7), 284.03134 (61), 285.03903 (100)✚✚66Myricetin 3-O-(2″-feruloyl)-rhamnoside8.45C_31_H_27_O_15_^–^639.13554639.132333.22178.99808 (3), 271.02359 (5), 316.02097 (100), 317.02841 (18)✚✚67Kaempferol 3-O-rhamnoside8.75C_21_H_19_O_10_^–^431.09837431.095702.67227.03384 (3), 255.02866 (9), 284.03122 (100), 285.03900 (95), 431.09540 (9)✚✚68Myricetin 3-O-(2″-*p*-coumaroyl)-rhamnoside8.76C_30_H_25_O_14_^–^609.12498609.121933.05178.99762 (5), 271.02374 (4), 287.01871 (4), 316.02087 (100), 317.02826 (23)✚✚69Quercetin 3-O-(2″-feruloyl)-rhamnoside8.78C_31_H_27_O_14_^–^623.14063623.138062.57151.00307 (8), 178.99770 (8), 271.0231 (8), 300.02597 (100), 301.03372 (91), 447.09412 (3)✚✚70Quercetin 3-O-hexuronide9.33C_21_H_17_O_13_^–^477.06746477.065861.61151.00308 (12), 178.99791 (9), 271.04382 (5), 301.03391 (100)✚–71Quercetin 3-O-(2″-galloyl)-rhamnoside9.95C_28_H_23_O_15_^–^599.10424599.101372.87151.00294 (8), 169.01347 (6), 178.99767 (8), 300.02606 (4), 301.03375 (100)✚✚*Flavonoid aglycones*







72Santin6.05C_18_H_15_O_7_^–^343.08233343.080172.16285.03925 (16), 299.05457 (5), 313.03403 (100), 328.05753 (52), 343.08069 (15)✚–73Myricetin7.26C_15_H_9_O_8_^–^317.03029317.028851.44151.00304 (75), 178.99777 (93), 227.03392 (37), 245.04449 (39), 255.02879 (43), 317.02881 (100)✚✚74Dihydrokaempferol7.30C_15_H_11_O_6_^–^287.05611287.054771.34107.01334 (9), 125.02382 (5), 135.04445 (77), 151.00298 (100), 171.04417 (5), 199.03915 (9)✚✚75Quercetin7.37C_15_H_9_O_7_^–^301.03538301.033921.46121.02889 (16), 151.00302 (100), 178.99782 (55), 273.03931 (10), 301.03397 (83)✚✚76Quercetin 3-methyl ether7.51C_16_H_11_O_7_^–^315.05103315.049231.80151.00238 (3), 242.02019 (4), 271.02213 (5), 300.02603 (100), 315.04932 (52)✚✚77Naringin7.73C_15_H_11_O_5_^–^271.06120271.059901.30107.01345 (11), 119.04965 (37), 151.00294 (100), 177.01817 (13), 227.03337 (5), 271.02338 (26)✚✚78Luteolin7.81C_15_H_9_O_6_^–^285.04046285.039031.43169.01361 (8), 241.07059 (4), 285.03879 (100)✚✚79Isorhamnetin8.19C_16_H_11_O_7_^–^315.05103315.049621.40271.02371 (8), 300.02606 (100), 315.04953 (8)✚✚80Apigenin8.19C_15_H_9_O_5_^–^269.04555269.044321.23117.03402 (2), 149.02419 (2), 151.00276 (4), 225.0518 (2), 269.04428 (100)✚–81Luteolin 3′,4′-dimethyl ether8.67C_17_H_13_O_6_^–^313.07176313.070341.42269.04468 (33), 283.02335 (33), 297.03931 (25), 298.04672 (100), 313.07043 (57)✚–82Taxifolin8.67C_15_H_11_O_7_^–^303.05103303.049731.30185.02388 (10), 213.01805 (14), 231.06580 (16), 257.04465 (37), 285.03912 (100)✚–83Genistein8.92C_15_H_9_O_5_^–^269.04555269.044221.33137.02353 (2), 241.04845 (2), 269.04401 (100)✚✚84Isokaempferide9.02C_16_H_11_O_6_^–^299.05611299.054681.43255.02927 (9), 284.03119 (100), 299.05472 (37)✚–85Kaempferol9.20C_15_H_9_O_6_^–^285.04046285.039141.32169.01358 (8), 257.04599 (4), 285.03864 (100)✚✚*Quinones*







86Juglanoside C gallate5.48C_23_H_23_O_12_^–^491.11950491.116652.85169.01335 (27), 211.02362 (25), 241.05040 (32), 271.04449 (100), 313.05499 (11), 473.10571 (10)✚✚87Isosclerone5.63C_10_H_9_O_3_^–^177.05572177.054820.90159.04437 (35), 177.05481 (100)✚✚88Juglanoside D5.80C_16_H_19_O_9_^–^355.10346355.101112.35175.03918 (100), 193.04956 (6), 235.05971 (3)✚✚89α-Hydrojuglone 4-hexoside5.96C_16_H_17_O_8_^–^337.09289337.091171.73175.03923 (100)✚✚90α-Hydrojuglone 4-hexoside gallate6.48C_23_H_21_O_12_^–^489.10385489.101062.79169.01343 (86), 174.03143 (76), 175.03922 (100), 211.02342 (28), 271.04471 (77), 313.05515 (75)✚✚91Jugnaphthalenoside A6.88C_23_H_19_O_12_^–^487.08820487.085512.69324.02600 (100), 325.03366 (51)✚✚921,4-Naphthalenedion hexoside7.17C_16_H_15_O_8_^–^335.07724335.075591.65135.04453 (22), 161.02367 (100), 335.07623 (11)✚–932-Methoxyjuglone7.51C_11_H_7_O_4_^–^203.03498203.033951.04174.03149 (31), 175.03922 (35), 203.03404 (100)✚✚942-Hydroxyjuglone7.51C_10_H_5_O_4_^–^189.01933189.018390.94161.02361 (14), 189.01833 (100)✚✚954,8-Dihydroxy-2-naphthalenecarboxylic acid hexoside9.16C_17_H_17_O_10_^–^381.08272381.080122.60174.03123 (9), 175.03949 (2), 218.02090 (100), 219.02734 (3)✚✚*Fatty acids*







962-Hydroxy-9,12,15-octadecatrienoic acid9.88C_18_H_29_O_3_^–^293.21220293.210271.93121.10181 (32), 171.10199 (36), 183.13818 (90), 211.13284 (14), 235.16917 (45), 275.20035 (100)✚✚97Linoleic acid10.81C_18_H_31_O_2_^–^279.23295279.230702.25279.23141 (100)✚✚98*cis*-Octadecenoic acid11.36C_18_H_33_O_2_^–^281.24860281.247061.54281.24719 (100)✚–99Palmitic acid11.95C_16_H_31_O_2_^–^255.23295255.231581.37255.23174 (100)✚✚*Other metabolites*







100Malic acid 0.53C_4_H_5_O_5_^–^133.01420133.013580.6271.01352 (35), 72.99274 (9), 89.02402 (7), 115.00318 (100), 133.01364 (48)✚✚101Citric acid0.80C_6_H_7_O_7_^–^191.01973191.018790.9457.03428 (7), 85.02903 (35), 87.00828 (50), 111.00817 (100), 129.01872 (7), 191.01768 (7)✚✚102Dihydrophaseic acid5.80C_15_H_21_O_5_^–^281.13945281.137891.55123.08092 (74), 171.11708 (100), 189.12749 (36), 201.12729 (39), 207.13826 (25), 237.14822 (75)✚✚103Bergaptol8.54C_11_H_5_O_4_^–^201.01933201.018311.02173.02385 (4), 201.01852 (100)✚–tR—retention time (min); Δ mDa—mean mass accuracy; NA—not available; ✚ stands for detected and—for not detected compound. In ethanol extracts of the leaves and the green husks, the total extracted substances and the total content of phenol and flavonoids are determined and given in Table 2. Significant (*p* < 0.05) higher total extracted substances, total phenolic, and flavonoid content were observed in the extract of leaves compared with the green husk. In order to evaluate the antioxidant activity of the extract from *J. nigra* leaves and green husks, the DPPH radical (1,1-diphenyl-2-picrylhydrazyl free radical) scavenging capacity was measured (Table 2). There was a significant difference (*p* > 0.05) in radical scavenging between the two extracts.
plants-13-01669-t002_Table 2Table 2Characterization of extracts from leaves and green husks of *J. nigra*.Characterization ParametersLeavesGreen HusksTotal extracted substances (mg cm^−3^)56.3 ± 0.4 *33.5 ± 0.1 *Total polyphenolic content (mg GA cm^−3^)10.66 ± 0.23 *6.24 ± 0.03 *Total flavonoid content (mg R cm^−3^)7.95 ± 0.65 *2.70 ± 0.45 *IC_50_ (µg cm^−3^)0.07 ± 0.010.11 ± 0.01Data are reported as means ± SD. * *p* < 0.05, compared pairs. GA—gallic acid; R—rutine.

Precise identification of the composition of the extracts obtained using specific conditions established by response surface methodology (RSM) and artificial neural network models with genetic algorithms (ANN-GAs) underscores the potential value of *J. nigra* leaves and husks extract. These compounds can be leveraged as valuable ingredients for dietary supplements or functional foods due to their diverse biological effects, making the quantification of individual compounds a crucial aspect to consider, as studied by Rajkovic et al., 2020 [18].

Electrochemical profiling of the *J. nigra* extracts showed different behaviors for the leaves and husk extracts. The results are summarized in Figure 2. As can be seen, leaves extract showed three times higher antioxidant capacity in comparison with husk extract. If we compare dominant peaks, it can be concluded that both extracts have almost the same potential peaks. This indicates that the structure of the polyphenolic compounds in both samples is the same or very similar. Rajković et al. reported that the dominant compounds in these samples are quercetin-based [18]. Taking into account that the electrochemical behavior of the quercetin gives three oxidation signals (1, 2, and 3 in Figure 2) at the same experimental parameters as used in this paper, it is conclusive that our results are fully in accordance with the literature data [19,20].

Interactions of the plant extracts with DNA provide important data about their behavior after human intake. Experiments were conducted as follows: 500 ppm of the free DNA electrochemical profile was recorded using CV, and after that, 0.1 mL of the extract was added. After 5 min of the incubation period, CV measurements were performed at the same experimental parameters. Results for the separate interaction of both nut extracts with double-stranded DNA are shown in Figure 3A,B. Based on the previous results, as we expected, the same amount of the leaves extract showed the highest interaction with the DNA. The second addition of 0.1 mL of the extract decreased DNA signals obtained from adenine (Ad) (Figure 3A,B) from the chain. Contrary to these results, green husk extract showed a lower decrease rate, and a total absence of the adenine oxidation peak was noted after the fifth addition of 0.1 mL of the extract, which is also in accordance with previous results. However, both extracts show high interaction rates with the DNA chain, indicating the high potential of these extracts in plant medicine.

Electrochemical profiling of the interactions between the leaves extract and the green husks extract was used for the estimation of the nature of their interactions with the DNA. As we can see, both extracts caused the same response with and without the presence of DNA. The increase in the amount of DNA, in the case of both extracts, is followedby slight shifts of the peak potentials to more negative values, around 40 mV for the green husks extract and 30 mV for the leaves extract, and this behavior is linked with the electrostatic interactions between extracts and the anionic phosphate backbone of DNA. Similar to these results, there are noted shifts in the peak potentials of the adenine toward more positive values (for 30–35 mV in the leaves study and around 50 mV in the green husks study), indicating the contribution of the intercalation mode of the interactions. In summary, we can describe and assign both modes of binding to the synergetic effects of intercalation and electrostatic mode.

The binding constant for each extract was calculated based on the study by Deepa et al. (2018), where the binding constant was calculated from the intercept of the plot of log (1/DNA) versus log (I/I_o_ − I) using the following equation:log (1/[DNA]) = log *K* + log (I/(I_o_ − I))
where *K* is the binding constant, and I_o_ and I are the oxidation peak currents of the extracts at the potential 0.5 V (I_500mV_) before and after 5 min of the interactions. The calculated values were for leaves extract 3.56 × 10^4^ M^−1^ and for green husks extract 0.76 × 10^4^ M^−1^. The obtained values are in the range of the binding constants for the polyphenolic compounds reported in the literature, suggesting great application potential for both extracts in the supplement industry and traditional medicine.

Using the obtained K values from the equation:C_b_/C_f_ = (I − I_DNA_)/I_DNA_
where I and I_DNA_ are the currents before and after interaction of the extracts and DNA, and C_b_ and C_f_ are concentrations of free and bonded DNA [21]. The calculated binding site sizes are 0.19 and 0.14 for the leaves and green husks extracts, respectively. A summary of the obtained values is given in Table 3. 

The significant content of phenolic and flavonoid compounds in both extracts [18,22] provides a chemical basis for their antioxidative activity. The previous studies relied on the capacity of scavenging DPPH free radicals when assessing the antioxidant abilities of tested compounds [10,22]. Therefore, in this study, the DPPH test was used to estimate the antioxidant activity of extracts from *J. nigra* leaves and green husks, respectively. The DPPH test showed that both extracts of *J. nigra* have high antioxidant activity.

All tested concentrations of ethanol leaves and green husks extracts were evaluated for their ability to protect against chromosome aberrations in peripheral human lymphocytes through the cytokinesis-block micronucleus (CBMN) assay. Amifostine WR-2721, used as a positive control, is a prodrug that can be transformed into an active sulfhydryl compound capable of scavenging radiation-induced free radicals and averting cell damage [23]. The lymphocyte cell culture was exposed to MMC, a clastogenic agent employed to assess cells’ susceptibility to chromosomal damage and cytotoxic effects. The isolated compounds were tested across different concentrations, and the frequencies and distribution of micronuclei (MN) in human lymphocytes were documented [24]. The results, including the comparison with negative and positive controls, are listed in Table 4. The alkylating agent MMC, when administered at a concentration of 0.2 μg/mL, led to a significant increase in MN frequency (30%) in contrast to the control cell cultures, while treatment with amifostine WR-2721 at a concentration of 1 μg/mL resulted in a significant decrease in MN frequency (24%) compared to the control cell cultures, as per the statistical analysis.

The statistical significance of the difference between the data pairs was evaluated by analysis of variance (one-way ANOVA), followed by the Tukey test. A statistical difference was considered significant at *p* < 0.01.

Compared with control groups, statistically significant difference *p* < 0.01.Compared with amifostine—WR 2721, statistically significant difference *p* < 0.01.Compared with mitomycine—C, statistically significant difference *p* < 0.01.

In the analysis, the tested *J. nigra* extracts demonstrated different effects on the frequency of micronuclei (MN) in cell cultures. For instance, the *J. nigra* green husks extract decreased the MN frequency by 64.5% at a concentration of 4 μg/mL, while its effects at concentrations of 2 and 6 μg/mL were slightly weaker, at 59% and 61.3%, respectively. By comparison, the *J. nigra* leaf extract exhibited slightly lower activity, reducing the MN frequency by 58.2% at a concentration of 4 μg/mL. At concentrations of 2 and 6 μg/mL, it also caused a significant decrease in MN frequency, with reductions of 52.1% and 55.5%, respectively. These effects were found to be more pronounced than those of amifostine (Table 2).

The impact of *J. nigra* extracts on cell proliferation was assessed using the cytokinesis-block proliferation index (CBPI). The calculated mean CBPI values for different concentrations of the extracts indicated an inhibitory effect on lymphocyte proliferation when compared to the positive control. Considering the direct dependence of MN expression on cell division, evaluating cell proliferation and cell death is essential for a comprehensive assessment of cell kinetics and MN frequencies.

This study revealed the impact of *J. nigra* at concentrations of 2.0, 4.0, and 6.0 μg/mL on reducing the frequency of MN in lymphocyte cell cultures. It is important to notice that the *J. nigra* extracts exhibited a substantial protective effect on human lymphocyte DNA, comparing to, and in some cases even better than, the synthetic antioxidant and cytoprotective agent amifostine.

## 3. Materials and Methods

### 3.1. Extract Preparation

*J. nigra* leaves and fruits were collected during the summer (2021) at Aleksinac locality, in the southeast region of Serbia. The voucher specimen was deposited at the Herbarium of the Department of Botany, University of Belgrade Faculty of Pharmacy, under the number 3906HFF. The leaves and green husk were dried in the air and grounded, thus obtaining plant material particles of an average size of 0.75 mm.

The grounded plant material was mixed with 70% (*v*/*v*) ethanol for 4 h under reflux at a solvent-to-solid ratio of 4:1. The suspension of plant particles in the solvent was cooled to room temperature, taken from the flask, and filtered under vacuum to separate the liquid extract from the solid residue.

### 3.2. The Electrochemical Measurements

Reagent-grade calf–thymus DNA (DNA) was supplied by Sigma Aldrich Chemical Co., St. Louis, MI, USA. The electrochemical measurements (cyclic voltammetry, CV) were performed using a potentiostat/galvanostat Autolab PGSTAT 302 N (MetrohmAutolab B.V., Utrecht, The Netherlands) controlled by Nova 2.0 software. Electrochemical measurements were conducted in three electrode glass cells (total volume of 20 mL) with a glassy carbon working electrode, Ag/AgCl electrode (3 M KCl) as the reference electrode, and a Pt wire as the counter electrode. Each potential reported in this paper is given against the Ag/AgCl/3 M KCl electrode at ambient temperature (25 + 1 °C). DNA-binding interactions were examined in two ways. First, several extracts (4 mg/mL) were added to the DNA solution at a concentration of 20 mg/mL to confirm the binding of the extracts to DNA. Second, to study the nature and constant for DNA/extract binding, a known amount of DNA was added to the extracts, and the current reduction was monitored.

### 3.3. Subjects

Venous blood samples were obtained using heparinized sterile vacutainers (Becton Dickinson, Franklin Lakes, NJ, USA) from four healthy, nonsmoking male volunteers who had not been exposed to chemicals, drugs, or other substances. A safety protocol concerning a blood-borne pathogen/biohazard was used. The volunteers gave permission to use their blood for the experiment. From each subject, two 5 mL aliquots of blood were obtained.

The study complied with the code of ethics of the World Medical Association (Helsinki Declaration of 1964, as revised in 2002). The blood samples were obtained at the Medical Unit in accordance with the current health and ethical regulations in Serbia (Law on Health Care, Serbia, 2019).

### 3.4. Characterization of Extracts

#### 3.4.1. Total Extracted Substances

The solvent obtained from the extract was then evaporated in a rotary vacuum evaporator until a half-solid residue was obtained, which was then dried at 60 °C to a constant weight. The dry residue represents the total extracted substances.

#### 3.4.2. Total Polyphenolic Content Determination

The total phenolic (TP) content in the extracts was determined by a VIS spectrophotometer (SPECO1) according to the Folin–Ciocalteu method using gallic acid as a standard. The tested extract (20 × 10^−3^ cm^−3^) and Folin–Ciocalteu reagent (1 cm^−3^) were placed in a 10 cm^−3^ volumetric flask. Aliquots (0.8 cm^−3^) of 7.5% aqueous Na_2_CO_3_ solution were added to the solution, and the reaction mixture was increased up to 10 cm^−3^ with distilled water. The absorbance of the mixture was measured after 30 min at 765 nm. For the quantification of total phenolic content, an external standard method was used with a standard of gallic acid at 765 nm (y = 1.7907x + 0.0244, R^2^ = 0.9983). The total phenolic content of the extract was expressed as gallic acid equivalents in mg per cm^−3^ extract (mg GA cm^−3^).

#### 3.4.3. Total Flavonoid Determination

The total flavonoid (TF) content in the extracts was determined with a VIS spectrophotometer (SPECO1) according to Al_2_O_3_ using rutin as a standard. Aliquots (0.8 cm^−3^) of 10% aqueous Al_2_O_3_ solution and 1 mol dm^−3^ aqueous solution CH_3_COOK (0.1 cm^−3^) were added to the tested extract (2 cm^−3^). The absorbance of the mixture was measured after 30 min at 430 nm. For the quantification of total flavonoid content, an external standard method was used with a standard of rutin at 430 nm (y = 7.2328x − 0.2286, R^2^ = 0.9919). The total flavonoid content of the extract was expressed as rutine equivalents in mg per cm^−3^ extract (mg R cm^−3^).

#### 3.4.4. DPPH Radical Scavenging Capacity

In order to evaluate the antioxidant activity of the extract from *J. nigra* leaves and green husks, the DPPH radical (1,1-diphenyl-2-picrylhydrazyl free radical) scavenging capacity was measured. Extracted solutions (0.3 cm^−3^, 20–200 μg cm^−3^) were incubated with DPPH solution (2.7 cm^−3^, 90 μmol dm^−3^) for 30 min in the dark, and afterwards the absorbance was measured at 517 nm using a VIS spectrophotometer (SPECO1). A blank control of the ethanol/water mixture was run in each assay. Inhibition of DPPH radical was calculated as a percentage (%) using the following equation:$$\mathrm{Scavenging}\;\mathrm{effect}\;(\%)\;=\left[\frac{A_{\mathrm{DPPH}}-A_{S}}{A_{\mathrm{DPPH}}}\right]\times 100$$
where A_S_ is the absorbance of the DPPH solution when the sample extract was added and A_DPPH_ is the absorbance of the DPPH solution. The extract concentration providing 50% inhibition (IC_50_) was calculated from the graph representing the dependence of the scavenging effect on the concentration of extracts. The IC_50_ value was obtained by a linear regression equation for the extract of leaves (y = 669.59x + 0.1, R^2^ = 0.9999) and green husks (y = 431.95x + 3.619, R^2^ = 0.9864). IC_50_ was expressed in µg cm^−3^. The measurements were performed in triplicate, and the data were presented as average ± standard deviations (SD).

#### 3.4.5. LC/MS Method for Metabolite Identification

LC-HRMS/MS (Thermo Scientific™ Vanquish™ Core HPLC system coupled to the Orbitrap Exploris 120 mass spectrometer, San Jose, CA, USA) was used to determine the metabolic profile of the extracts.

The liquid chromatography system was equipped with a Hypersil GOLD™ C18 analytical column (50 × 2.1 mm, 1.9 μm particle size), thermostated at 40 °C. The injection volume was 5 μL, and the flow rate was constant at 300 μL/min. The compounds of interest were eluted with ultrapure water + 0.1% formic acid (A) and acetonitrile (MS grade) + 0.1% formic acid (B): 5% B in the first min; 5–95% B from 1 to 10 min; 95% B from 10 to 12 min; 5% B until 15 min. The used characterization technique was previously optimized by the researchers [25].

The Orbitrap Exploris 120 mass spectrometer was equipped with a heated electrospray ionization (HESI-II) source operating in negative ionization modes. Full-scan MS was monitored from 100 to 1500 *m*/*z* with an Orbitrap resolution set to 60,000 FWHM, while data-dependent MS^2^ experiments were conducted at an Orbitrap resolution of 15,000 FWHM. Normalized collision energy was set to 35% with an isolation width of 1.5 *m*/*z*. The dynamic exclusion time was set to 10 s with exclusion from a specific scan after two occurrences, and the intensity threshold was set to 1 × 10^5^.

LC/MS dates were evaluated using R Studio (version 2023.09.1, build 494) software. Peak picking was performed using the enviPick R package, and peak correspondence across samples was performed using the density method available in the xcms R package [25]. The identification of the metabolites was performed based on their chromatographic behavior and HRMS/MS^2^ data by comparison with standard compounds, when available, and literature data providing a tentative identification [25,26,27,28,29,30,31,32,33]. Data acquisition was carried out with the Xcalibur^®^ data system (Thermo Finnigan, San Jose, CA, USA).

### 3.5. In Vitro Cytokinesis-Block Micronucleus (MN) Assay

The lymphocyte cultures were treated with investigated ethanol extracts of the *J. nigra* leaves and husks at concentrations of 2, 4, and 6 μg/mL. Untreated cell culture served as a blank control. One cell culture containing the known clastogenic agent mitomycin C (MMC; Calbiochem, Merck Chemicals, Darmstadt, Germany, ≥95% purity by HPLC) (0.2 μg/mL, in phosphate buffer) alone was used as a negative control. One cell culture containing Amifostine WR-2721 (98%, S-2[3-aminopropylamino]-ethylphosphothioic acid) at 1.0 μg/mL), (Marligen-Biosciences, Ijamsville, MD, USA) was used as a positive control. They were added to the cultures 25 h after phytohaemaglutinin (PHA) stimulation and left until harvest. All cultures were incubated in a thermostat at 37 °C. Treatment with the investigated ethanol extracts lasted for 19 h, after which all cultures were rinsed with a pure medium, transferred into 5 mL of fresh RPMI 1640 medium (RPMI 1640 Medium + GlutaMAX + 25 mM HEPES; Invitrogen-Gibco-BRL, Vienna, Austria), and incubated for an additional 72 h. Approximately 2 × 10^6^ blood lymphocytes were set up in 5 mL of RPMI-1640 medium supplemented with 15% calf serum and 2.4 μg/mL of phytohaemaglutinin (Invitrogen-Gibco-BRL, Waltham, MA, USA). One hour after initiating the cell stimulation, investigated ethanol extracts (three concentrations) were added to the samples.

The incidence of spontaneously occurring MN in control samples was scored. For MN preparation, the cytokinesis-block method of Fenech and Morley [3] was used with some modifications, as described by Stankovic et al. [34]. At least 1000 binucleated (BN) cells per sample were scored, registering MN according to the criteria of Countryman and Heddle [35] and Fenech and Morley [3]. The slides were air-dried and stained with alkaline Giemsa 2% (Sigma-Aldrich, Vienna, Austria). At least 1000 binucleated (BN) cells per sample were scored, registering MN according to the criteria of Countryman and Heddle and Fenech and Morley [3,35]. The effects of the investigated complexes on cell proliferation were estimated by the cytokinesis-block proliferation index (CBPI), calculated as suggested by Surralles et al. [36].

### 3.6. Statistics and Index Calculations

Statistical analysis was performed using the Origin software package, version 7.0. The statistical significance of differences between data pairs was evaluated by analysis of variance (one-way ANOVA), followed by the Tukey test. A difference was considered significant at *p* < 0.01. The results are presented as the percentage of change compared to the control.

## 4. Conclusions

In conclusion, ethanolic extracts from *J. nigra* leaves and green husks show high antioxidant power, which is based on the presence of polyphenolic compounds. At the same concentration level, leaves extract possesses higher activity and more diversity in the presented polyphenolic compounds, connected with the stronger ability of the binding to DNA and a five-fold higher contract. This was confirmed by calculating binding site size, where a value of around 50% higher was obtained for this extract. The nature of the binding to DNA for both extracts was found to belong to the synergy of intercalative and electrostatic modes. We found that the lower concentration of *J. nigra* exerts a beneficial effect on lymphocyte cell culture by decreasing the frequency of MN. Since the number of micronuclei serves as an indicator of DNA damage, these results indicate that *J. nigra* protects DNA and decreases lipid peroxidation in lymphocytes, mostly induced by superoxide anion radicals. Free radicals disturb cellular homeostasis by peroxidation of membrane lipids, oxidation of proteins, base damage, and adduct formation in DNA, which ultimately leads to cell death if the damage is beyond cell repair capacity. Our results provide evidence of the protective effects *J. nigra* exerts on cytogenetic damage in human lymphocytes treated *in vitro*. These extracts have great potential to be promising candidates to be used as new therapeutic agents and can be considered excellent antioxidant sources in the pharmaceutical industry.

Therefore, our future research will be focused on the investigation of the radioprotective effects of *J. nigra* extracts after cancer radionuclide therapy.

## Figures and Tables

**Figure 1 plants-13-01669-f001:**
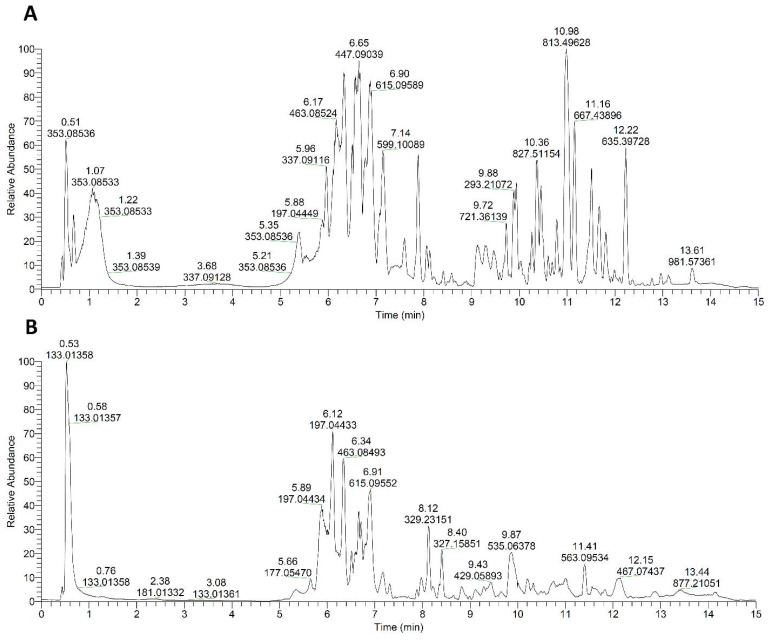
Base peak chromatograms of walnut (**A**) leaf and (**B**) husk ethanolic extracts.

**Figure 2 plants-13-01669-f002:**
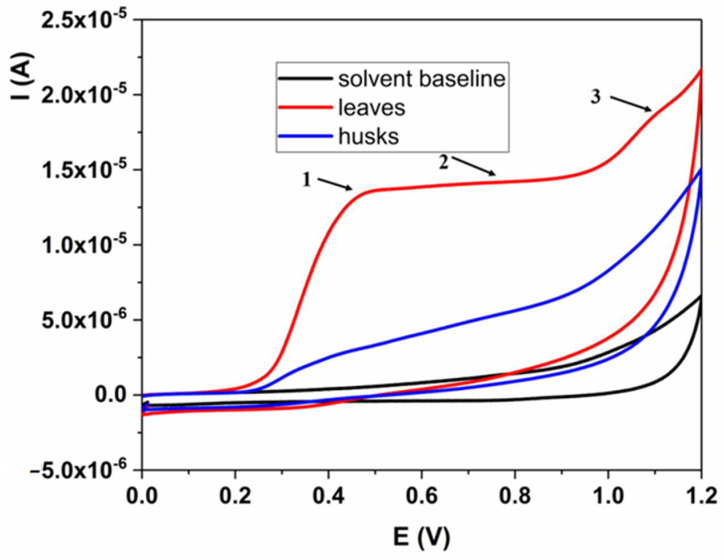
Electrochemical profiling of the green husks and leaves extract.

**Figure 3 plants-13-01669-f003:**
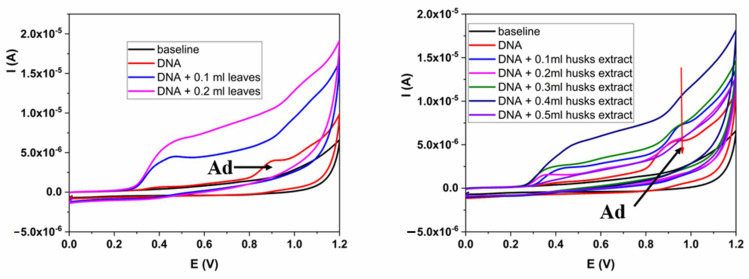
(**A**) DNA interaction with leaves extract; (**B**) DNA interaction with green husks extract.

**Table 3 plants-13-01669-t003:** Binding constant, binding site size, and type of interaction for the extracts.

Type of Extract	Binding Constant	Binding Site Size	Type of Interaction
Leaves	3.56 × 10^4^ M^−1^	0.14	Electrostatic/intercalation
Green husks	0.76 × 10^4^ M^−1^	0.19	Electrostatic/intercalation

**Table 4 plants-13-01669-t004:** Frequency of MN, cytokinesis-block proliferation index, distribution of MN per cell, and frequency of MN measurement in cell cultures of human lymphocytes treated with various concentrations of extract of *J. nigra* (leaves and green husks).

Conc.	MN/1000	% Bn Cell	MN/Bn	CBPI	Frequency
µg/mL	Bn cell	with MN	cell		of MN %
Control	26.85 ± 0.23	2.23 ± 0.08	1.19 ± 0.03	1.67 ± 0.03	100%
Amifos.—1.0	20.38 ± 0.58 ^a^	1.88 ± 0.04	1.21 ± 0.03	1.64 ± 0.02	75.9% (−24.1%)
MMC—0.2	34.82 ± 0.63 ^a,b^	3.36 ± 0.13	1.12 ± 0.03	1.71 ± 0.06	129.68% (+29.68%)
leaves—2.0	12.86 ± 0.43 ^a,b,c^	1.22 ± 0.02	1.10 ± 0.02	1.70 ± 0.08	47.89% (−52.11%)
leaves—4.0	11.23 ± 0.35 ^a,b,c^	1.27 ± 0.16	1.12 ± 0.07	1.61 ± 0.03	41.82% (−58.18%)
leaves—6.0	11.96 ± 0.38 ^a,b,c^	1.09 ± 0.01	1.21 ± 0.05	1.61 ± 0.03	44.54% (−55.46%)
green husks—2.0	10.99 ± 0.54 ^a,b,c^	1.15 ± 0.02	1.18± 0.03	1.66 ± 0.03	40.93% (−59.07%)
green husks—4.0	9.54 ± 0.32 ^a,b,c^	0.75 ± 0.07	1.17 ± 0.04	1.75 ± 0.06	35.53% (−64.47%)
green husks—6.0	10.39 ± 0.50 ^a,b,c^	0.86 ± 0.09	117 ± 0.05	1.60 ± 0.02	38.70% (−61.30%)

**MN/1000 Bn cells**—incidence of micronuclei in 1000 binucleated cells. **% Bn cells** witch micronuclei. **MN/Bn cells**—incidence of micronuclei in binucleated cells. **CBPI**—cytokinesis-block proliferation index.

## Data Availability

Data are contained within the article.

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
