# Peer review of "Chemical Composition and Protective Possibilities of Juglans Nigra Leaves and Green Husks Extracts: DNA Binding and Micronucleus Assay in Human Lymphocytes"

_plants, 2024, doi:10.3390/plants13121669_

Round 1

Reviewer 1 Report

Comments and Suggestions for Authors

This work investigates the interaction of two leaf and husk walnut extracts with DNA using cyclic voltammetry and evaluates their potential protective effect on human lymphocytes. The experimental strategy appears appropriate, the results are presented clearly, and the conclusions seem to align with them.

In my opinion, the main weakness of the study is the determination of phenols and flavonoids using colorimetric methods, which are not very specific, while UHPLC-MS is only used for qualitative analysis of the samples.

Some details in the text need attention:

Line 143: How can it be explained that there are no differences in the DPPH of the two extracts despite the leaf extract having a much higher phenol content?

Line 148: Please specify the meaning of “RSM” (Response Surface Method?) and “ANN-GA” (Artificial Neural Network-Genetic Algorithm?) models

Line 160: The authors indicate the consistency of their results with previous works, noting that the samples are based on quercetin derivatives, which have three oxidation signals. However, in the voltammogram in Figure 2, these three signals are not evident. Could you, please point out these signals in the figure?

Line 172-173: Please consider pointing out the peak of adenine oxidation in Figure 3A to clarify the whole experiment.

Line 210-213: “In addition, to evaluate the antioxidant activity of extracts from leaves and green green husks, the DPPH radical scavenging capacity was measured. The DPPH scavenging capacity of leaves and green husks extract expressed as IC50 was 0.07 ± 0.01 and 0.11 ± 0.01 μg cm-3, respectively. There was significant difference (p<0.05) in radical scavenging between the two extracts.” The statement in this paragraph seems contradictory to the data in Table 2 and to the assertion in lines 143-144, "There was not significant difference (p>0.05) in radical scavenging between the two extracts."

Line 221-222: “All the individual compounds were evaluated for their ability to protect against chromosome aberrations in peripheral human lymphocytes through the CBMN assay.” Please explain what do you mean with “All the individual compounds” in this sentence.

Line 222: Please specify the meaning of the abbreviation “CBMN” (cytokinesis-block micronucleus?)

Comments on the Quality of English Language

English language could be improve

Author Response

Dear Editor,

Thank you so much for giving us an opportunity to revise our manuscript, and thus improve our work according to the reviewers’ valuable comments. We have carefully considered all reviewers’ questions and all the weak points of our work and have significantly changed our manuscript accordingly. The list of changes has been written below, and inserted changes in the text of the re-submitted manuscript are being highlighted. We hope that the revised version of the manuscript will satisfy yours and the reviewers’ requirements for publication in the Journal.

Referee Comments:

Referee 1:

This work investigates the interaction of two leaves and husk walnut extracts with DNA using cyclic voltammetry and evaluates their potential protective effect on human lymphocytes. The experimental strategy appears appropriate, the results are presented clearly, and the conclusions seem to align with them.

In my opinion, the main weakness of the study is the determination of phenols and flavonoids using colorimetric methods, which are not very specific, while UHPLC-MS is only used for qualitative analysis of the samples.

We are grateful to the reviewer for its time and efforts and positive comments of our work. We revised manuscript according to these comments and we hope that this improvement, based on reviewer comments will significantly improve final version of our work.

Some details in the text need attention:

Q1: Line 143: How can it be explained that there are no differences in the DPPH of the two extracts despite the leaf extract having a much higher phenol content?

Thank you very much for the comment. The values ​​for IC50 were misspelled in Table 2. We apologize for the typo as the data provided in the paper in line 210-213 (your comment Q5) is correct, so we have corrected Table 2.

We revised the sentence as follows (lines 143-144): There was significant difference (p>0.05) in radical scavenging between the two extracts.

Q2: Line 148: Please specify the meaning of “RSM” (Response Surface Method?) and “ANN-GA” (Artificial Neural Network-Genetic Algorithm?) models

A2: Thanks for the comment. We revised the sentence as follows (Line 148):

Precise identification of the composition of the extracts obtained using specific conditions established by response surface methodology (RSM) and artificial neural network models with genetic algorithms (ANN-GA) underscores the potential value of J. nigra leaves and husks extracts.

Q3: Line 160: The authors indicate the consistency of their results with previous works, noting that the samples are based on quercetin derivatives, which have three oxidation signals. However, in the voltammogram in Figure 2, these three signals are not evident. Could you, please point out these signals in the figure?

A3: Thank you for noticing the lack of clear labeling of the peaks to which the discussion relates. In the revised version of the paper, the mentioned peaks are marked in Figure 2 and noted in the text at the appropriate place.

Figure 2. Electrochemical profiling of the green husks and leaves extract.

Q4: Line 172-173: Please consider pointing out the peak of adenine oxidation in Figure 3A to clarify the whole experiment.

A4: Thank you for your comment about the lack of clear labeling of the adenine peak. In the revised version of the paper, the peak is marked (Ad) both in the text of the paper and in Figure 3.

Figure 3. A) DNA interaction with leaves extract; B) DNA interaction with green husks extract.

Q5: Line 210-213: “In addition, to evaluate the antioxidant activity of extracts from leaves and green green husks, the DPPH radical scavenging capacity was measured. The DPPH scavenging capacity of leaves and green husks extract expressed as IC50 was 0.07 ± 0.01 and 0.11 ± 0.01 μg cm-3, respectively. There was significant difference (p<0.05) in radical scavenging between the two extracts.” The statement in this paragraph seems contradictory to the data in Table 2 and to the assertion in lines 143-144, "There was not significant difference (p>0.05) in radical scavenging between the two extracts."

A5: Thanks for the comment. We explained ahead in Q1. As the results are inserted in Table 2, this paragraph is deleted.

Q6: Line 221-222: “All the individual compounds were evaluated for their ability to protect against chromosome aberrations in peripheral human lymphocytes through the CBMN assay.” Please explain what do you mean with “All the individual compounds” in this sentence.

A6: Thanks for your kind reminders. We revised the sentence as follows [Ln221--222]

All tested concentrations of ethanol leaves and green husks extracts were evaluated for their ability to protect against chromosome aberrations in peripheral human lymphocytes through the cytokinesis-block micronucleus (CBMN) assay.

Q7: Line 222: Please specify the meaning of the abbreviation “CBMN” (cytokinesis-block micronucleus?)

A7: Thank you very much for the reminder. We have made revisions accordingly.

Reviewer 2 Report

Comments and Suggestions for Authors

In this study, the authors investigated the chemical composition and some of the biological properties of  Juglans nigra leaves and green husks extracts. The topic of the manuscript is interesting. However, I have some remarks and recommendations as follows

1. Has the LC-HRMS method been previously optimized and validated? Please add relevant information.

2. Why was DPPH chosen for antioxidant properties testing? DPPH admittedly is widely used but has disadvantages in addition to its advantages. DPPH reactions are very sensitive to the reaction medium, such as: water and solvent, pH, light exposure, dissolved oxygen. Since the ionization of phenols – and consequently the reaction rates – are highly influenced by solvent composition and pH, the DPPH assay may not be suitable for ranking antioxidant compounds and natural extracts.

3. I suggest that the authors write more extensively about what therapeutic uses the plant extracts studied may have.  What are the future perspectives?

4. Any study limitations should be presented and clearly explained. 

5. There are some typos in the text. The text of the manuscript should be carefully checked.

Author Response

In this study, the authors investigated the chemical composition and some of the biological properties of  Juglans nigra leaves and green husks extracts. The topic of the manuscript is interesting. However, I have some remarks and recommendations as follows

Thank you for your time and effort. We appreciate all comments and conclusions. In the revised version, we have accepted all valuable suggestions and tried to improve our manuscript. All changes are highlighted in the revised version.

Q1. Has the LC-HRMS method been previously optimized and validated? Please add relevant information.

A1: Thank you for your contribution, which will significantly contribute to the understanding of the experimental part of the work and be of significant benefit to other researchers. The used characterization technique was previously optimized by the researcher who conducted the mentioned tests and as such was taken from the previous study. In the revised version of the paper, this was clearly emphasized and a corresponding sentence was added to the Experiential part that corresponds to this measurement.

Q2: Why was DPPH chosen for antioxidant properties testing? DPPH admittedly is widely used but has disadvantages in addition to its advantages. DPPH reactions are very sensitive to the reaction medium, such as: water and solvent, pH, light exposure, dissolved oxygen. Since the ionization of phenols – and consequently the reaction rates – are highly influenced by solvent composition and pH, the DPPH assay may not be suitable for ranking antioxidant compounds and natural extracts.

A2: We are well aware of both the conceptual and technical issues that limit the use and compromise the validity of the three most commonly used assays – TEAC/ABTS•+, DPPH and ORAC. Since we have the most experience and knowledge with the DPPH test, in this paper we decided to use it for testing antioxidant properties.

According to literature data, one of the most popular colorimetric tests for evaluating the radical scavenging capacity of plants and extracts is the 1,1-diphenyl-2-picrylhydrazyl (DPPH) test. This method is accurate, easy to perform and economical, providing screening of general antioxidant activity and is based on the stable and synthetic radical, DPPH(A.V. Badarinath, K. Mallikarjuna Rao, C. Madhu Sudhana Chetty, S. Ramkanth, T.V.S. Rajan, K. Gnanaprakash, Int. J. PharmTech. Res. 2 (2010) 1276–1285. Gonçalves, A. C., Bento, C., Jesus, F., Alves, G., & Silva, L. R. (2019). Sweet Cherry Phenolic Compounds: Identification, Characterization, and Health Benefits. Studies in Natural Products Chemistry, 31–78. doi:10.1016/b978-0-444-64179-3.00002-5)

In our further research, we will use other tests to test antioxidant properties and compare the obtained results with the results of the DPPH test.

Q3: I suggest that the authors write more extensively about what therapeutic uses the plant extracts studied may have.  What are the future perspectives?

A3: We agree with this comment. As we presented in the introduction what the extracts of individual parts of the Juglans nigra are used for, we added the sentence in the conclusion:

Therefore, our future research will be focused on the investigation of the radio-protective effects of J. nigra extracts after cancer radionuclide therapy.

Q4: Any study limitations should be presented and clearly explained.

A4: We thank the editors for this helpful comment.

In this paper, the DPPH test, as a chemical test, was used for the preliminary assessment of the antioxidant activity of the extract. This test has several advantages and disadvantages compared to other free radical scavenging tests. It also has limitations regarding its applicability to complex biological systems and diverse mechanisms of action.

However, the goal of this work is to use other tests to obtain data on the effect of compounds from leaves and green husks extracts on biological systems, i.e. on DNA, as well as the in vitro protective effect of extracts against mitogen-induced chromosomal damage in human lymphocytes.

Q5: There are some typos in the text. The text of the manuscript should be carefully checked.

A5: Thank you very much for the reminder. We have made revisions accordingly.

Round 2

Reviewer 1 Report

Comments and Suggestions for Authors

The manuscript has been duly modified in accordance with the reviewer's comments.

    4o Comments on the Quality of English Language

The manuscript has been duly modified in accordance with the reviewer's comments.

Reviewer 2 Report

Comments and Suggestions for Authors

The authors responded to my comments and revised the manuscript.  I appreciate their responses. The paper could be published in the current form.